# Development of Electrically Conductive Thermosetting Resin Composites through Optimizing the Thermal Doping of Polyaniline and Radical Polymerization Temperature

**DOI:** 10.3390/polym14183876

**Published:** 2022-09-16

**Authors:** Kohei Takahashi, Kazuki Nagura, Masumi Takamura, Teruya Goto, Tatsuhiro Takahashi

**Affiliations:** 1Graduate School of Organic Materials Science, Yamagata University, 4-3-16 Johnan, Yonezawa 992-8510, Japan; 2Open Innovation Platform, Yamagata University, 4-3-16 Jonan, Yonezawa 992-8510, Japan

**Keywords:** electrically conductive, thermosetting resin, polyaniline

## Abstract

This work developed an electrically conductive thermosetting resin composite that transitioned from a liquid to solid without using solvents in response to an increase in temperature. This material has applications as a matrix for carbon fiber reinforced plastics. The composite comprised polyaniline (PANI) together with dodecyl benzene sulfonic acid (DBSA) as a liquid dopant in addition to a radical polymerization system made of triethylene glycol dimethacrylate with a peroxide initiator. In this system, micron-sized non-conductive PANI particles combined with DBSA were dispersed in the form of conductive nano-sized particles or on the molecular level after doping induced by a temperature increase. The thermal doping temperature was successfully lowered by decreasing the PANI particle size via bead milling. Selection of an appropriate peroxide initiator also allowed the radical polymerization temperature to be adjusted such that doping occurred prior to solidification. Optimization of the thermal doping temperature and the increased radical polymerization temperature provided the material with a high electrical conductivity of 1.45 S/cm.

## 1. Introduction

Carbon fiber reinforced plastics (CFRPs) prepared by infusing liquid monomers into carbon fiber fabric sheets with subsequent thermosetting are typically lightweight but exhibit high strength. For these reasons, CFRPs have been used as structural materials in aircraft as replacements for various metals. However, aircraft are sometimes subject to lightning strikes [1] and therefore CFRPs that provide lightning strike protection (LSP) are of considerable importance. Specifically, while the metal components of aircraft are electrically conductive and so are not significantly damaged by lightning strikes [2], CFRPs can be severely deteriorated as a result of the decomposition of the CFs and/or the resin [3]. These effects can occur as a consequence of the very low electrical conductivity of typical CFRPs in the thickness direction (which results from the non-conductive matrix resin) and of the Joule heating generated by the large electric current induced by a lightning strike.

Currently, LSP is afforded to the CFRP components of aircraft by applying metal mesh sheets to the surfaces of these items as well as at the interfaces between individual CF layers. However, this increases the mass of the CFRP parts and also the complexity of the fabrication process [4]. For these reasons, the development of electrically conductive thermosetting resins has received considerable attention.

One approach to creating conductive thermosetting resins is to add electrically conductive filler to these materials. There are many types of conductive fillers such as metals and carbon materials, and many studies have been done to add them to thermosetting resins and CFRP [5,6,7,8,9].

When added in the form of particles (either micro- or nano-sized) or as individual molecules, these polymers provide channels for current flow. As an example, the electrically conductive polymer polyaniline (PANI) is often used as a filler for CFRPs because this substance is lightweight, readily synthesized, inexpensive and highly stable compared with other such polymers [10,11,12]. PANI typically exists in a non-conductive state referred to as emeraldine base (EB) but can be transformed to a conductive emeraldine salt (ES) by the addition of an acid. The acid acts as a dopant and changes the electrical state of the nitrogen atoms contained in the quinoid structure of the macromolecule. PANI is generally not soluble or meltable due to the rigid molecular structure imparted by the benzene rings along each chain as well as the strong molecular interactions in this polymer. However, PANI molecules or nanoparticles can be dispersed in various solvents or liquid resins in conjunction with a liquid acid dopant having a molecular weight of 200 to 500. Typical dopants are do-decyl benzene sulfonic acid (DBSA) and camphor sulfonic acid [10,11,12].

The thermal properties and LSP characteristics of CFRPs can be improved by incorporating a dispersed PANI/dopant composite in the thermosetting resin [13,14,15]. However, achieving a suitable degree of electrical conductivity requires a high concentration of this composite, which in turn increases the viscosity of the liquid resin (based on the large surface area of the conductive particles, which leads to significant interparticle friction and interactions). This high viscosity hinders the penetration of the resin into the CF fabric.

To overcome the above difficulty, we have previously proposed and studied the use of PANI/DBSA in conjunction with thermal doping [16,17,18,19,20]. In this process, DBSA (which is a liquid at room temperate) is mixed with micron-sized PANI particles formed from aggregated nanoparticles during the initial fabrication stage. Doping does not occur at this relatively low temperature and this combination produces a low-viscosity dispersion. Rather, the liquid DBSA is only able to penetrate micron-sized aggregated particles and nano-sized particles. With increasing temperature, doping effects appear due to the penetration of the DBSA into the nano-sized particles [16] and the resulting DBSA-doped PANI nano-particles can be dispersed in thermosetting resins [20].

This process is unique in that the DBSA acts as a thermal dopant and also as an initiator for the cationic polymerization of the monomer. The infusion of a liquid thermosetting resin containing PANI/DBSA into stacked layers of CF fabric sheets followed by hot pressing has been found to provide good dispersion of PANI/DBSA nanoparticles or molecules within the individual CF fibers. This process, therefore, generates PANI/DBSA connections (that is, achieves thermal doping) in the thermosetting resin to produce a CFRP having high electrical conductivity in the thickness direction [21]. These materials have thus exhibited excellent LSP properties [22,23,24]. The CFRP is intended to be applied to parts of aircraft wings and structural materials for the fuselage [3,4]. The CFRP is lightweight because it does not have a metal mesh, so it has the advantages of improving fuel efficiency and reducing carbon dioxide emissions.

Unfortunately, as the room-temperature polymerization of the resin is promoted by the DBSA, which is difficult to inhibit, the viscosity of the mixture increases [25].

This uncontrolled increase in viscosity can be avoided by using different compounds as the dopant and the initiator based on polymerization with a radical initiator. This system also allows the polymerization initiation temperature to be controlled by careful selection of the radical initiator. On this basis, a new design comprising a mixture of PANI, a methyl methacrylate monomer including phosphoric acid as the dopant (P-2M), and a peroxide (Peroxide butyl E) was proposed by Santwana et al. This system provided a constant viscosity of 2500 mPa·s by inhibiting polymerization at room temperature, along with a conductivity of 0.5 ± 0.18 S/m and a flexural modulus of 2.6 ± 0.13 GPa [26] for the pure resin without CFs. When this material was used as the CFRP matrix, the conductivity of the material in the thickness direction was 0.14 ± 0.02 S/cm, which is potentially suitable for LSP [27].

However, this newly developed radical polymerization thermosetting resin system resulted in reduced conductivity, possibly because the material underwent polymerization before the thermal doping process could be completed, since the hour half-life decomposition temperature of Peroxide butyl E has a relatively low value of 119 °C. In addition, the doping effect of phosphoric acid appeared to be less than that of sulfonic acid. On this basis, it would evidently be beneficial to use sulfonic acid as the thermal dopant while controlling the radical polymerization temperature based on using an optimal peroxide.

The present study examined thermal doping by DBSA in conjunction with thermosetting by radical polymerization and assessed the means of increasing the conductivity of the finished product. Experimental trials were carried out using PANI, DBSA, methacrylate (as a bi-functional monomer) and a peroxide. The effect of using DBSA for thermal doping in conjunction with different mixers and the effect of promoting radical polymerization with peroxides having different decomposition temperatures on the conductivity of the resin were investigated. The purpose of the study was to develop a high conductivity thermosetting resin system via the optimization of thermal doping (i.e., by decreasing the thermal doping temperature) and radical polymerization (i.e., by increasing the thermosetting temperature).

## 2. Materials and Methods

### 2.1. Raw Materials

Triethylene glycol dimethacrylate (TEGDMA; Tokyo Chemical Industry Co., Ltd., Tokyo, Japan) was used as the liquid monomer in this work (Figure 1a) together with di-t-hexyl peroxide (Peroxide H; Figure 1b) and di-t-butyl peroxide (Peroxide B; Figure 1c; NOF Corp., Tokyo, Japan). These compounds had initiation temperatures (that is, hour half-life temperatures) of 136.2 and 144.1 °C respectively. PANI Emeraldine Base (PANI-EB; Regulus Co., Ltd., Tokyo, Japan), a non-conductive polymer (Figure 1d), was also employed, together with DBSA (Kanto Chemical Co., Inc., Tokyo, Japan) as a liquid compound capable of acting as a dopant in response to a temperature increase (Figure 1e). TEGDMA was employed as the monomer because each molecule contains two reactive acrylate groups that are available for radical polymerization, and so this compound is suitable for the formation of a three-dimensional network in response to initiation by oxygen radicals (-O•) produced by the Peroxide H and Peroxide B as a result of thermal decomposition.

### 2.2. Preparation and Evaluation of PANI/DBSA/TEGDMA Composites

The PANI/DBSA/TEGDMA composites were prepared as liquid pastes using the procedures summarized in Figure 2a,b. In this work, either a centrifugal mixer or a bead mill was used to prepare the composite so as to maintain the original PANI particle size or to produce smaller PANI particles, respectively. The intent was to identify the effect of particle size on the thermal doping process.

(a)Centrifugal mixer method

DBSA and TEGDMA (both liquids) were added to PANI-EB powder to obtain a PANI/DBSA/TEGDMA mass ratio of 18/32/50. The mixture was stirred by hand and then transferred into a bottle and agitated using a centrifugal mixer (ARE-310, THINKY Corp., Tokyo, Japan) at 2000 rpm for 5 min. This procedure generated a liquid paste referred to herein as CM-PANI/DBSA/TEGDMA.

(b)Bead mill method

The DBSA, TEGDMA and PANI-EB were combined in the same manner as described above and stirred first manually and then by a magnetic stirrer for 20 min. The resulting liquid paste was then processed in a bead mill (EasyNano RMB-01 AIMEX Co., Ltd., Tokyo, Japan) at 1000 rpm for 20 min using 160 g of zirconia spheres each having a diameter of 0.5 mm. After milling, the zirconia spheres were removed to give the product, referred to herein as BM-PANI/DBSA/TEGDMA.

(c)Thermal analysis and optical microscopy observations

Doping of the PANI-EB with DBSA in both the CM-PANI/DBSA/TEGDMA and BM-PANI/DBSA/TEGDMA was monitored using differential scanning calorimetry (DSC; DSC Q-200, TA Instruments Japan Inc., Tokyo, Japan) at a heating rate of 5 °C/min from 40 to 190 °C. These trials allowed an evaluation of the effect of the different particle sizes in these specimens. Thermal doping of DBSA into PANI-EB without a solvent is known to reduce the size of PANI particles and aggregates and this effect was evaluated using polarized optical microscopy (BX50, Olympus Corp., Tokyo, Japan) to observe the CM-PANI/DBSA/TEGDMA and BM-PANI/DBSA/TEGDMA while heating the materials from 30 to 150 °C at a rate of 5 °C/min. In situ optical microscopy observations of heated samples were also performed. In these trials, each liquid paste sample was placed between a slide glass and a cover glass to give a constant specimen thickness, after which the material was heated while performing in situ observations.

### 2.3. Preparation of Composites Including Radical Initiators

Four samples were prepared using combinations of the two different mixing methods and two different peroxides, as summarized in Table 1. In two of these samples, 5 mol% of Peroxide H (based on the moles of TEGDMA in the formulation) was added to the liquid composite. Following this, samples 1 and 2 were processed using the centrifugal mixer at 2000 rpm for 5 min. In the same manner, samples 3 and 4 were made with Peroxide B. Samples 1, 2, 3, and 4 were evaluated by thermal analysis under the same conditions as Section 2.2. (c). Microscopic observations were also evaluated under the same heating conditions, but caution should be exercised as the sample thickness differs from Section 2.2. due to contraction or expansion of the composite during thermosetting.

### 2.4. Conductivity Measurements

Samples 1, 2, 3, and 4 prepared in Section 2.3. were heat treated and then measured for conductivity. The liquid composites were poured into metal containers (25 mm in diameter, 1 mm in thickness) and processed using a hot press device (Mini test press, Toyo Seiki Seisaku-sho, Ltd., Tokyo, Japan) at 135 °C and 2 MPa for 1 h to induce radical polymerization. The cured sample produced by hot pressing was allowed to cool to room temperature (approximately 25 °C), and the electrical conductivity was then measured. The conductivity of each specimen was subsequently determined using the four probe method in conjunction with a Loresta-GP MCP-T600 apparatus (Dia Instrument Co., Ltd., Yokohama, Japan). It should be noted that this device actually measured surface resistivity values, from which electrical conductivity data were obtained based on the equation
ρv = ρs × t,(1)
where ρv is the volume resistivity (Ω·cm), ρs is the surface resistivity (Ω/sq) and t is the specimen thickness (cm) and
σ = 1/ρv.(2)

To ensure accurate measurements of the conductivity, the surface skin layer on each sample (which comprised an ultrathin layer of non-conductive resin) was carefully removed by hand prior to each measurement.

## 3. Results

### 3.1. Effect of the Thermal Doping Temperature

Thermal doping was used to introduce the liquid dopant DBSA into the PANI without using solvents, based on raising the temperature of these materials. The driving force for this doping was the polar interaction between the PANI and DBSA, while the liquid state of the DBSA allowed it to penetrate into the PANI particles. These factors, in turn, were affected by the temperature as well as the PANI particle size and morphology. The latter two parameters could be modified by changing the conditions applied during the polymerization of the PANI. Because of the lack of a solvent system, a temperature increase was required to allow penetration of the liquid dopant, meaning that this was a thermal doping process. This type of system has been frequently reported in the literature [10,11]. It is also known that thermal doping is an endothermic phenomenon that can be followed using DSC.

In the present work, it was important to characterize the thermal doping phenomenon for a simple system without a peroxide initiator. Figure 3 presents the DSC curves obtained from the two liquid PANI/DBSA/TEGDMA composites without peroxide (prepared according to the process detailed in Section 2.2). Figure 3a provides data for the CM-PANI/DBSA/TEGDMA and indicates an exotherm with a peak at 103 °C attributed to the doping of DBSA into the PANI particles [10,11,19]. In contrast, the data in Figure 3b obtained from the BM-PANI/DBSA/TEGDMA sample indicate an exothermic phenomenon at 92 °C. Despite their different locations, the exothermic peaks generated by the two specimens are almost identical. Considering that bead milling produced smaller PANI particles than centrifugal mixing, the decrease in the peak temperature from 103 to 92 °C is attributed to accelerated penetration of the liquid DBSA into the smaller PANI particles.

Figure 4 presents the resulting images of the PANI/DBSA/TEGDMA samples. Here, the transparent yellow areas are the liquid regions consisting of TEGDMA and DBSA while the black areas are the PANI-EB particles and the greenish areas are the DBSA-doped PANI (that is, PANI in the emeraldine salt state (PANI-ES)). At 40 °C, both the CM-PANI/DBSA/TEGDMA and BM-PANI/DBSA/TEGDMA contained aggregates with sizes in the range of 100–200 μm, although the aggregates in the former were generally larger. Thus, although bead milling reduced the particle size, all materials contained aggregates.

In Figure 4a, the black regions representing initial aggregates in the CM-PANI/DBSA/TEGDMA are seen to occupy more of the image as the temperature is increased, suggesting that the PANI particles became better dispersed as the DBSA penetrated into the TEGDMA. It should be noted that this behavior did not occur in a mixture composed of only PANI and TEGDMA. In addition, the image acquired at 135 °C indicates that the sample had a slight greenish coloration, suggesting the presence of doped PANI molecules. These images together with the DSC curves confirm that the PANI particles were dispersed during the doping process, such that the aggregates expanded and connected networks were formed. These phenomena would be expected to enhance the conductivity of the material.

Figure 4b presents similar images obtained from the BM-PANI/DBSA/TEGDMA at 40, 100 and 135 °C It is evident that the aggregates became more closely connected with increasing temperature so that the boundaries between particles disappeared. A greenish color appeared at 100 °C corresponding to the peak in the DSC curve for this sample, and became more pronounced at 135 °C Again, this greenish color is attributed to the doping of the PANI molecules.

Figure 5 provides a series of diagrams that summarize the optical microscopy observations. Initially, the CM-PANI/DBSA/TEGDMA and BM-PANI/DBSA/TEGDMA contained aggregates of larger and smaller PANI particles, respectively. At 40 °C, the aggregates were not connected. Increasing the temperature caused these aggregates to increase in size while reducing the sizes of the individual PANI particles, resulting in the connection of the aggregates and inducing electrical conductivity.

### 3.2. Control of the Thermosetting Temperature

Figure 6 summarizes the DSC curves for the four different composites (that is, samples 1 through 4). The exotherms at approximately 100 °C in these plots are ascribed to thermal doping. The exotherms appearing at 150 °C in Figure 6a,b (samples 1 and 3) are related to the radical polymerization of the TEGDMA by Peroxide H. While those at 160 °C in Figure 6a,b (samples 2 and 4) are due to reaction with Peroxide B. Because the hour half-life temperature of Peroxide B is higher than that of Peroxide H, the thermosetting reaction temperatures for samples 2 and 4 were higher than those for samples 1 and 3. These data, therefore, demonstrate the possibility of tuning the thermosetting temperature in the presence of PANI/DBSA by varying the peroxide. Figure 6b shows double exotherm peaks in both curves above 160 °C that corresponds to the radical polymerization, while the exotherms in Figure 6a are single peaks. These results are ascribed to inhibition of the thermosetting reaction based on the trapping of oxygen radicals generated by the peroxides in the smaller particles of PANI produced via bead milling. This effect, in turn, promoted vaporization of the TEGDMA, which has a boiling point of 155 °C. This effect did not occur during radical polymerization of the actual samples at 135 °C.

### 3.3. Optimization of Factors Affecting Conductivity

In this work, two key factors (thermal doping as discussed in Section 3.1 and thermosetting as discussed in Section 3.2) were adjusted so as to optimize the conductivity of the proposed new composites. Figure 7 summarizes the conductivity data for the four different composites. Samples 3 and 4, both of which were prepared by bead milling, exhibited higher conductivities, in good agreement with the effective formation of networks of smaller PANI particles. In addition, the conductivity values of Sample No. 2 and No. 4 are approximately an order of magnitude higher than those of Sample No. 1 and No. 3. This difference is attributed to the higher solidification temperature obtained when using Peroxide B. Overall, these data confirm the synergistic effect of decreasing the thermal doping temperature based on bead milling and increasing the thermosetting temperature based on the use of Peroxide B. Optimization of these two factors so as to reduce the thermal doping temperature and increase the radical polymerization temperature provided a high electrical conductivity of 1.45 S/cm.

## 4. Discussion

Figure 8 summarizes the causes of the large conductivity differences obtained in Section 3.3. It was clear that the thermal doping of PANI/DBSA dominated the conductivity of the composite. Conductivity increases as heat doping progresses, and conductivity decreases as heat doping is hindered. In the case of this radical polymerization system, it was clarified that the thermal decomposition temperature of the peroxide has a great influence on the progress of heat doping.

When peroxide H with a low thermal decomposition temperature was used (Samples 1 and 3), a crosslinked structure was formed by radical polymerization during the heat doping. And it is considered that the change of PANI-ES and the dispersion of PANI were not sufficient, and the electrical conductivity of the composite became low.

When peroxide B, which has a high thermal decomposition temperature, was used (Samples 2 and 4), a crosslinked structure was formed by radical polymerization after the heat doping proceeded. It is considered that sufficient changes in PANI-ES and PANI dispersion occurred, and the electrical conductivity of the composite increased by one order of magnitude. In addition, by lowering the heat doping temperature performed in Section 3.1, even with the same peroxide, the conductivity is higher when the change to PANI-ES and PANI dispersion is more likely to occur. 

Since it is found that the thermal decomposition temperature has a greater effect than the heat doping temperature, it is necessary to focus on the investigation of peroxides for further improvement of the conductivity and strength of the composite in the future. [26,27,28,29]

## 5. Conclusions

A new and highly conductive thermosetting resin was developed. This material is a liquid paste at room temperature but undergoes solidification upon heating. This material is made of an electrically conductive polymer produced by thermal doping of PANI with DBSA and a thermosetting resin that undergoes radical polymerization, comprising TEGDMA with a peroxide. Thermal doping of this composite was examined by DSC and in situ optical microscopy in conjunction with processing by centrifugal mixing or bead milling prior to adding a peroxide. Using the bead mill was found to lead to more effective network formation among aggregates that enhanced the conductivity of the material in response to temperature increases. DSC data demonstrated that the temperature associated with thermal doping was reduced after processing by bead milling as a consequence of the more effective penetration of DBSA into smaller PANI particles. The radical polymerization temperature was successfully tuned by varying the peroxide. Optimization of these factors provided a composite with an elevated conductivity of 1.45 S/cm. The high electrical conductivity of this composite is similar to or better than that of PANI-containing thermoset resins used in lightning strike protection CFRP studies [12,13,17,18,21]. Furthermore, in the case of a radical polymerization system, there is an advantage that DBSA and monomers do not react at room temperature and the viscosity does not increase. It can be expected to fabricate larger sizes of CFRP such as aircraft structural materials. However, since the epoxy resin used for CFRP, which is a general structural material, is a cationic polymerizable thermosetting resin, there is a disadvantage that the material of this research cannot be applied immediately. In the future, for the application of CFRP as a structural material, we will investigate the properties including strength and heat resistance as well as the fabrication of electrically conductive CFRP.

## Figures and Tables

**Figure 1 polymers-14-03876-f001:**
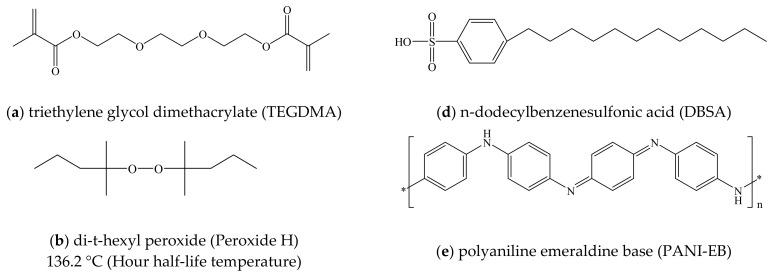
Molecular structures of (**a**) TEGDMA, (**b**) di-t-hexyl peroxide, (**c**) di-t-butyl peroxide, (**d**) DBSA and (**e**) PANI-EB.

**Figure 2 polymers-14-03876-f002:**
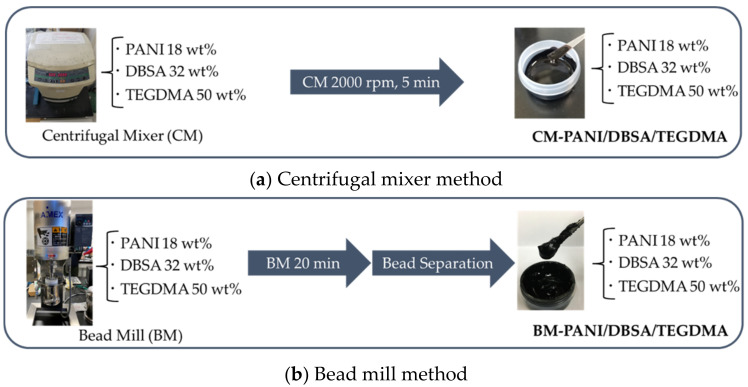
Procedures used to produce PANI/DBSA/TEGDMA composites in conjunction with (**a**) centrifugation and (**b**) bead milling.

**Figure 3 polymers-14-03876-f003:**
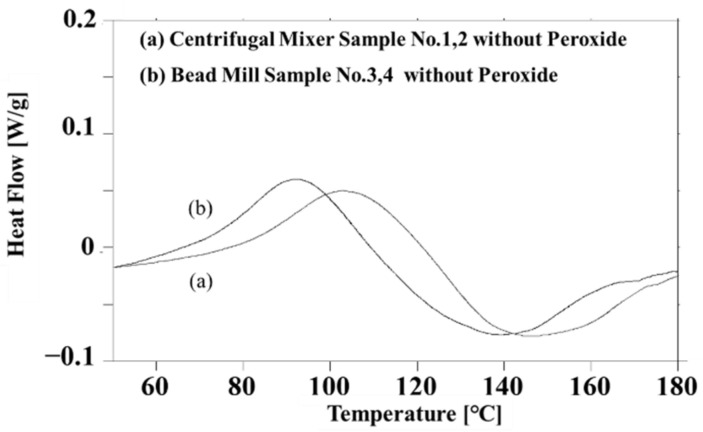
DSC data obtained from “Centrifugal mixer samples 1 and 2 without peroxide” and “Bead mill samples 3 and 4 without peroxide.

**Figure 4 polymers-14-03876-f004:**
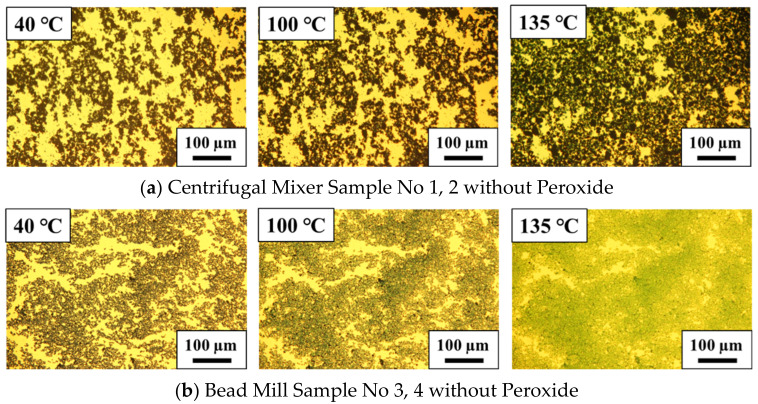
Optical microscopy images of (**a**) centrifugal mixer samples 1 and 2 and (**b**) bead mill samples 3 and 4. All samples were made without a peroxide.

**Figure 5 polymers-14-03876-f005:**
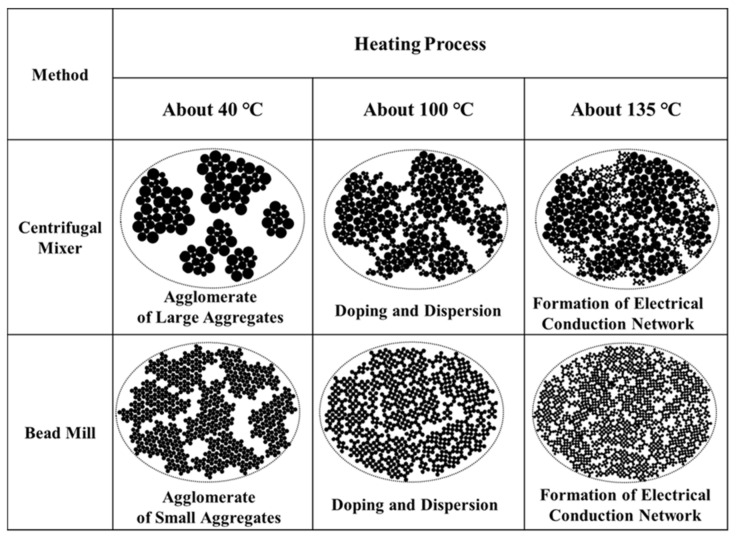
Diagrams summarizing the effects of the mixing method on thermally-induced dispersion and doping in the PANI/DBSA system.

**Figure 6 polymers-14-03876-f006:**
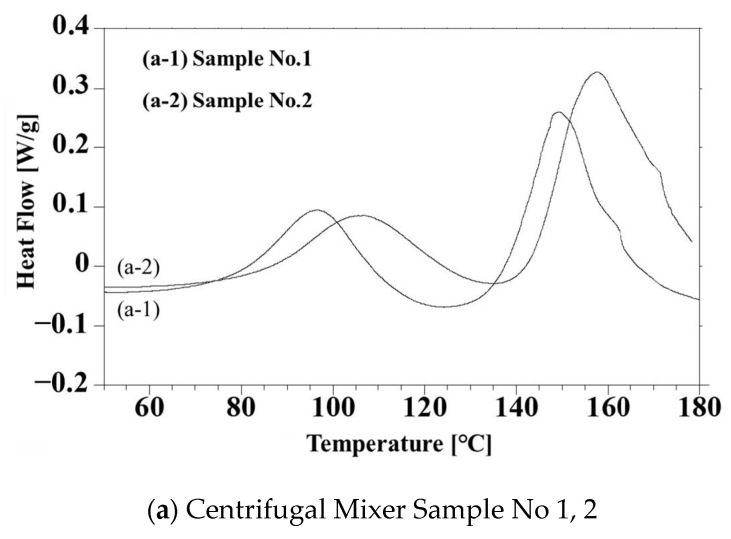
DSC data obtained from (**a**) Samples 1 and 2 and (**b**) Samples 3 and 4. Samples were processed using peroxides.

**Figure 7 polymers-14-03876-f007:**
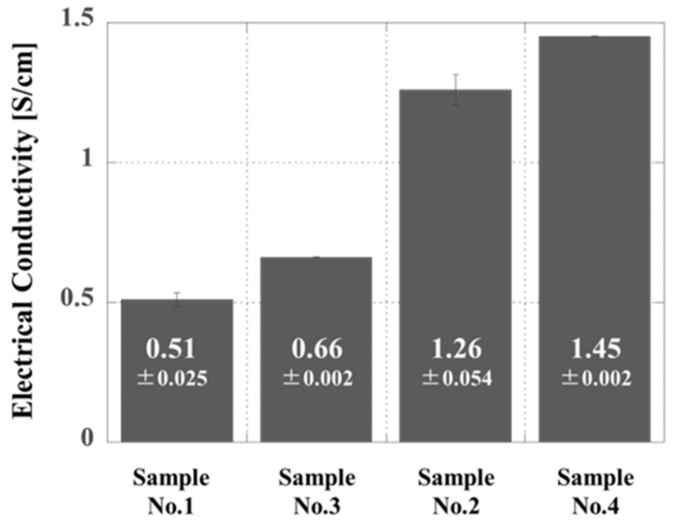
Electrical conductivities of samples 1 and 3 and samples 2 and 4 after heating. The cured sample produced by hot pressing was allowed to cool to room temperature (approximately 25 °C), and the electrical conductivity was then measured.

**Figure 8 polymers-14-03876-f008:**
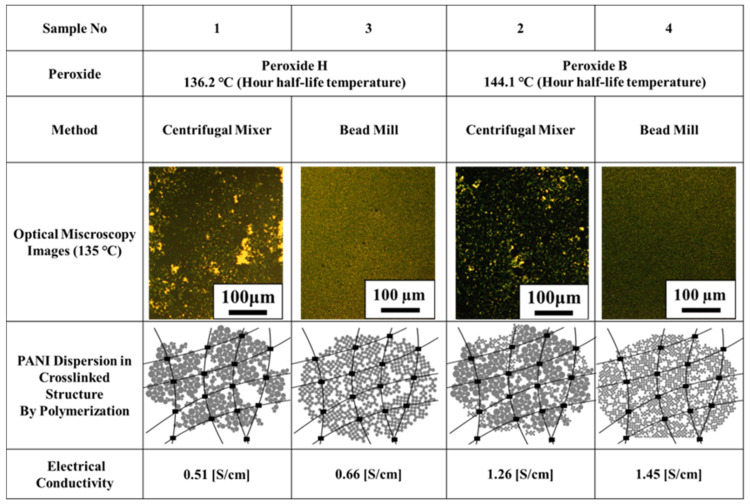
Summary of the thermal doping of PANI/DBSA and differences in the thermal decomposition temperatures of the peroxides and effects on conductive network formation.

**Table 1 polymers-14-03876-t001:** Compositions of test specimens (wt%).

Sample No	PANI	DBSA	TEGDMA	Mixing Method	Peroxide (5 mol% of TEGDMA)
1	18	32	50	Centrifugal mixer (Figure 2a)	Peroxide H
2					Peroxide B
3				Bead mill (Figure 2b)	Peroxide H
4					Peroxide B

## Data Availability

The raw data presented in this study are available on request from the corresponding author.

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
