# Peer review of "Development of Electrically Conductive Thermosetting Resin Composites through Optimizing the Thermal Doping of Polyaniline and Radical Polymerization Temperature"

_polymers, 2022, doi:10.3390/polym14183876_

Round 1
Reviewer 1 Report
This work is presenting the synthesis and subsequent thermal and electrical testing of polyaniline-based polymers, also studying the effect of the presence of peroxide in the process. The motivation is well-established. I would have some questions and suggestions for the authors before publication:
Line 89: A middle dot is missing between "mPa" and "s".
Line 97: Degrees symbol is missing (°C).
Figure 1: Please use lowercase fonts in the names of the molecules for consistency with the text, e.g. "Triethylene glycol dimethacrylate".
Line 130: Please remove the full stop at "Figures. 2(a)".
Tables in pages 4 and 5:
1) The table in page 4 corresponds to thes samples without peroxide, though it includes the peroxides?
2) That table also needs to be numbered and get a title, as well as the numbering of the other tables adjusted accordingly.
3) Best would be to call the samples series of the two tables differently for better clarity, because both those and the ones of the current table 1 are "1,2,3,4". In the discussion they are called with the suffix with and without peroxide, however there are instances where that is not the case, for example in line 245 of Figure 8. This might be confusing.
Figure 2: Please add a space between value and unit at "2000rpm", "5min", and "20min".
Page 5: Something is wrong with the headers of sections 2.3 and 2.4 (they are identical).
Page 5, conductivity measurements: Were the samples dried (perhaps under vacuum?) after polymerization to prevent influence of residual peroxide in the subsequent testing?
Line 178: A middle dot is missing between "Ω" and "cm".
Lines 259 and 260: Full stop missing.
Figure 7: At which temperature was conductivity measured? Especially given the DSC response, it might change quite a lot depending on the temperature?
Would there be any optical miscroscopy images of the samples with peroxide available? (similar to Figure 4). It would be key to see differences with the samples without peroxide, as well as verifying the model presented in Figure 8.
Reviewer 2 Report
This work developed an electrically conductive thermosetting resin composite that transitioned from a liquid to solid without using solvents in response to an increase in temperature. This material has applications as a matrix for carbon fiber reinforced plastics. The investigations are interesting and the paper could be published after revision. * Optimization of thermal doping temperature and the increased radical polymerization temperature provided a material with a high electrical conductivity of 1.45 S/cm. The conductivity should be compared with other conductive plastics. * Why the peroxides as di-t-hexyl peroxide (Peroxide H) and di-t-butyl peroxide were chosen for the polymerization. * Why5 mol% of Peroxide H was used ?How the amount was optimized ? *Advantages and disadvantages of the developed polymeric composites should be compared in conclusions with that of other similar products, which are described in literature. * A practical application of the developed composite as structural materials which is suitable as replacements for various metals should be demonstrated.Author Response
Please see the attachment

Round 2
Reviewer 1 Report
Thanks to the authors for addressing the remarks and adding further into in the introduction and conclusions.
Points 1/a-3 and 1/a-8 are still to be amended.
Point 3: Please add that clarification on sample preparation also in the manuscript in the first paragraph of section 2.4.
Point 4: Please add the mentioned measurement protocol to the manuscript both in the first paragraph of section 2.4 as well as (the temperature) in Figure 7.
Point 5: Understandable, though it would greatly support the schematic of Figure 8, even if the image quality is worse.
Reviewer 2 Report
If editor and other reviewers agree that the paper is suitable for this journal I would recommend the paper for publication after the revision.
Author Response
Thank you for your comment.